# Learning General Policies for Planning through GPT Models

**Primary Keywords:** *Learning*

## Abstract

Transformer-based architectures, such as T5, BERT and GPT, have demonstrated revolutionary capabilities in Natural Language Processing. Several studies showed that deep learning models using these architectures not only possess remarkable linguistic knowledge, but they also exhibit forms of factual knowledge, common sense, and even programming skills. However, the scientific community still debates about their reasoning capabilities, which have been recently tested in the context of automated AI planning; the literature presents mixed results, and the prevailing view is that current transformer-based models may not be adequate for planning. In this paper, we address this challenge differently. We introduce a GPT-based model customised for planning (PLANGPT) to learn a general policy for classical planning by training the model from scratch with a dataset of solved planning instances. Once PLANGPT has been trained for a domain, it can be used to generate a solution plan for an input problem instance in that domain. Our training procedure exploits automated planning knowledge to enhance the performance of the trained model. We build and evaluate our GPT model with several planning domains, and we compare its performance w.r.t. other recent deep learning techniques for generalised planning, demonstrating the effectiveness of the proposed approach.

## Introduction

Pre-trained Language Models and Large Language Models (LLMs) employing attention mechanisms represent the state-of-the-art in Natural Language Processing (NLP) tasks. Starting from the Transformer architecture (Vaswani et al. 2017), and then with BERT (Devlin et al. 2019) and GPT (Radford and Narasimhan 2018), LLMs have achieved SOTA results in different NLP tasks, such as machine translation and summarisation. Although these models can capture some forms of general knowledge of real-world facts (e.g., historical facts, geography, and medicine) (Petroni et al. 2019; Jiang et al. 2020), basic common sense (Geva et al. 2021) and programming skills (Wang et al. 2021), they have limited reasoning capabilities, such as logical inference or AI planning. In particular, some studies suggest that LLMs cannot generate valid plans to solve automated planning problems using simple prompting strategies or fine-tuning (Valmeekam et al. 2023, 2022; Arora and Kambhampati 2023). However, the recent work on Plansformer (Pal-lagani et al. 2023) shows that it is possible to fine-tune a pre-trained language model with planning instances from simple planning domains obtaining promising results.

In the context of *generalised planning* (e.g., (Hu and De Giacomo 2011; Srivastava, Immerman, and Zilberstein 2008, 2011)), several works have demonstrated that deep learning models can learn a *general policy*, i.e. a strategy employed to solve a set of different problems in a given planning domain (Groshev et al. 2018; Ståhlberg, Bonet, and Geffner 2022a,b). An example of general policy for the BLOCKSWORLD domain is to place all blocks on the table and then stack them in the desired position. However, deep learning approaches for learning general policies are often used to guide the search, without directly tackling the problem of finding a valid plan. Moreover, they are often limited to image-based domains (Groshev et al. 2018) or have logical restrictions, such as the one in (Ståhlberg, Bonet, and Geffner 2022a) which limits the approach to the two-variable fragment of first-order logic.

In this paper, we investigate generalised planning through transformer-based architectures. We introduce a GPT-based model customised for planning (PLANGPT) to learn a general policy for classical planning by training the model from scratch with a dataset of solved planning instances. Once PLANGPT has been trained for a domain, it can be used to generate a solution plan for an input problem instance in that domain. Our training procedure exploits automated planning knowledge to enhance the performance of the trained model. In particular, to prevent overfitting in training, we design and exploit an early-stopping technique validating the planning performance of the model while being trained. We build and evaluate PLANGPT with several planning domains, and we compare its performance with respect to other recent deep learning techniques for generalised planning, demonstrating the effectiveness of the proposed approach.

The paper is organised as follows: first, we discuss related work and provide background information. Then, we describe the preprocessing phase, the training datasets, and how the GPT models are designed and trained. Finally, we present the experimental results and draw our conclusions.

## Related Work

Recently several researchers have leveraged pre-trained LLMs to address planning and evaluate their reasoning capa-

bilities. For instance, the studies in (Valmeekam et al. 2022, 2023) assess how pre-trained GPT models (GPT-3.5 and GPT-4) can generate plans. They do not modify the original GPT model, but simply query it by exploiting some few-shot prompting frameworks to tackle planning tasks across various benchmark domains, such as BLOCKSWORLD and LOGISTICS. Their results highlight poor performance in generating valid plans that satisfy the problem goal.

In (Arora and Kambhampati 2023) a fine-tuned GPT-2 model is evaluated for the BLOCKSWORLD domain. Starting from a model pre-trained on text data, they built a generator of single actions, which are also progressively verified by another GPT-2 model fine-tuned on this specific task, reporting promising performance (about $60\%$ of problems solved over a custom-made test set of 200 instances). Another fine-tuning approach, called Plansformer, is proposed in (Pallagani et al. 2022, 2023) starting from a Code-T5 model (Wang et al. 2021) fine-tuned with solved problems in several planning domains. Although Plansformer obtains almost $90\%$ valid plans, these results are not directly comparable to those in (Arora and Kambhampati 2023), since they are obtained using two different test sets.

Researchers also exploited different types of deep learning architectures in the context of learning general policies. Toyer et al. (2018; 2020) use a custom feed-forward neural network to represent states and actions as neural layers and obtain the following action in probabilistic planning. The work in (Groshev et al. 2018) employed Convolutional Neural Network (CNN) to plan in the Sokoban domain with the current state represented as an image. To deal with propositional domains without image representations, they implemented a Graph Neural Network (GNN) that solves the travelling salesperson problem by building a plan selecting the next city to visit at each iteration. Similarly, Ståhlberg et al. (2022a; 2022b) adopt a GNN to tackle various benchmark domains. Given the current state, the system computes all the states reachable by applicable actions and selects the state with the best heuristic value estimated by the GNN, choosing it as the new current state; the procedure is repeated until a state is reached where goal is satisfied. An important difference of this approach w.r.t our work is that we use deep learning to directly generate the next actions rather than to evaluate heuristic values of states. Moreover, the expressive power of GNNs is restricted to the two-variable fragment of first-order logic with counting quantifiers (C2) (Ståhlberg, Bonet, and Geffner 2022a,b), while CNNs can only elaborate states encoded as images (Groshev et al. 2018).

Instead of fine-tuning or prompt-engineering a pre-trained model, in our work, we build a custom transformer-based model (trained from scratch) to learn a general policy for planning. This approach is followed by many other researchers to solve tasks with a high level of complexity, such as genetics-related challenges (Jumper et al. 2021) and the generation of programming code (Wang et al. 2021). In automated planning, Serina et al. (2022) trained a BERT model from scratch with a dataset of plans across various domains to solve the plan recognition problem of predicting missing actions from an observed partial plan. However, this method is not applicable for learning general policies as it ignores information about the initial states and goal of the class of problems handled by the policy.

## Background

### Classical Planning and General Policies

We assume that the reader is familiar with the standard planning language PDDL for representing deterministic, fully observable planning problems, of which here we present the most relevant elements following the formalisation given in (Ståhlberg, Bonet, and Geffner 2022a).

A classical planning problem is a pair $P = (D, I)$ where $D$ is a planning domain and $I$ is a problem instance. The planning domain $D$ contains a set of predicate symbols $p$ and a set of action schemas with preconditions and effects given by atoms $p(x_1, ..., x_k)$ where each $x_i$ is an argument of the schema. The problem instance is a tuple $I = (O, Init, Goal)$ where $O$ is a (finite) set of objects names $c_i$, and $Init$ and $Goal$ are sets of ground atoms $p(c_1, ..., c_k)$ representing the initial state and the goal of the problem. A classical problem $P = (D, I)$ encodes a state model $S(P) = (S, s_0, S_G, Act, A, f)$ where each state $s \in S$ is a set of ground atoms from $P$, $s_0$ is the initial state $Init$, $S_G$ is the set of goal states $s \in S$ such that $Goal \subseteq s$, $Act$ is the set of ground actions in $P$, $A(s)$ is the set of ground actions whose preconditions are true in $s$, and $f$ is the transition function so that $f(a, s)$ for $a \in A(s)$ represents the state resulting from applying action $a$ to state $s$. An action sequence $a_0, ..., a_n$ is applicable in $P$ if $a_i \in A(s_i)$ and $s_{i+1} = f(a_i, s_i)$, for $i = 0, ..., n$, and it is a plan if $s_{n+1} \in S_G$. The cost of a plan is assumed to be given by its length, and a plan is optimal if there is no shorter plan.

Generalised planning studies the representation and computation of general policies to solve multiple problems in the same planning domain (e.g., (Hu and De Giacomo 2011; Srivastava, Immerman, and Zilberstein 2008, 2011)). A general policy can be defined as a function $\pi(s, Goal)$ providing the next action in $Act$ to apply given the current state $s \in S$ and the goal of the problem instance $Goal$. A policy $\pi$ solves a set of classical planning instances for the same domain $D$ if each of these instances $I = (O, Init, Goal)$ is solved by the sequence of actions $\pi(s_0, Goal), ..., \pi(s_n, Goal)$, where $s_0 = Init$ and $Goal \subseteq s_{n+1}$.

Several approaches to generalised planning based on deep learning, including PLANGPT, adopt this representation of general policy (Groshev et al. 2018; Toyer et al. 2018, 2020). An alternative method is to define a value function in which the policy selects the successor state with the minimum value given the current state, goals and action, as in (Ståhlberg, Bonet, and Geffner 2022a,b).

### Generative Pretrained Transformer

GPT (Radford and Narasimhan 2018), which stands for Generative Pretrained Transformer, is a transformer-based architecture (Vaswani et al. 2017) originally designed to analyse sequences of elements in natural language processing (NLP) tasks. In the NLP context, these sequences are sentences or documents divided into *tokens* (words or part

of words). In our planning context, as detailed in the following sections, we will consider sequences of fluents and actions derived from the initial states, the goals, and the solution plans of planning problems.

The division of the sequence into tokens is performed by a probabilistic algorithm called *tokeniser*, which, through an analysis of the training set, also collects all different tokens into a vocabulary of size $v$. Typically, given a sequence of tokens in input, a GPT model is trained to generate another sequence in response, such as the translation of a sentence into another language, an answer to a question, or, in our case, a sequence of actions solving a planning instance. This generation is done one token at a time. In the following, we describe how the GPT architecture works considering the $i_{th}$ token $t_i$ in a sequence of $N$ tokens.

First, the model encodes the input token $t_i$ into an embedding vector $E_i \in \mathbb{R}^e$. This operation is performed through an embedding matrix $E \in \mathbb{R}^{v \times e}$ that embeds each word in a numeric vector of length equal to the embedding size $e$. Then, the model sums $E_i$ with the positional encoding vector $P_i \in \mathbb{R}^e$ obtaining the vector $I_i \in \mathbb{R}^e$. After the embedding phase, the first block processes the input through multiple masked self-attention mechanisms, typically called *heads* and other neural network layers.

In a self-attention mechanism (without considering masking), the model projects $I_i$ into three new representations called *key* ($K_i \in \mathbb{R}^d$), *query* ($Q_i \in \mathbb{R}^d$) and *value* ($V_i \in \mathbb{R}^d$) multiplying it with three weight matrices $W_k \in \mathbb{R}^{e \times d}$, $W_q \in \mathbb{R}^{e \times d}$ and $W_v \in \mathbb{R}^{e \times d}$, where $d$ is the dimension of the attention vectors. Then, the model calculates the dot-product between $Q_i$ and all $K_j$ in the sequence, where $K_j$ is the key vector of the $j_{th}$ token in the sequence. The model concatenates the results and applies the softmax function, obtaining a vector $A_i \in \mathbb{R}^N$, called *attention weight*. Each element of the attention weight $a_{i,j}$ ideally represents the interaction between the $i_{th}$ token and the $j_{th}$ token of the sequence. The head then calculates a new representation of $t_i$, $R_i \in \mathbb{R}^d$, by averaging the value representations of all tokens in the sequence multiplied by the respective attention weight. Whereas the traditional self-attention calculates the attention weights considering all tokens in a sequence, in the *masked-self attention* mechanism, for the $i_{th}$ token, only the tokens with a position $j \leq i$ are considered and the attention weights $a_{i,j}$ with $j > i$ are set to 0.

Each block of GPT applies $n$ heads at the same time (multi-head attention). In order to create a single representation of the context, the model concatenates the result of each head, obtaining a vector $M_i \in \mathbb{R}^q$, where $q = d \times n$, which is passed to a feed-forward layer that transforms $M_i$ into the new output vector $O_i \in \mathbb{R}^e$. Then there is a feed-forward layer and two residual connections with layer normalisation which ends a GPT block. The overall task of these blocks is to compute a more informative representation of each token of the same size $e$. The output of a block is the input of the next one. After the last block, the output of the last block is multiplied by a weight matrix, obtaining a vector of length $N$. A softmax layer then turns this vector into a probability distribution among all the tokens in the vocabulary. Finally, GPT outputs the token with the highest probability.

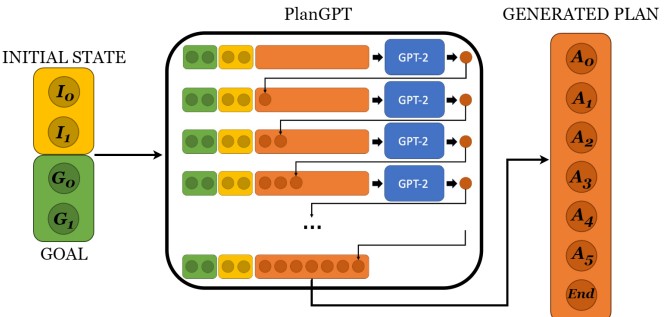

Figure 1: Architecture of PLANGPT and example of input/output illustrated for a planning problem with two fluents in the initial state ($I_0$ and $I_1$) and two fluents forming the goal ($G_0$ and $G_1$). PLANGPT generates as output the plan ($A_0$, $A_1$, ..., $End$).

The overall process of GPT begins generating the first token; then this is added to the input and GPT continues generating the second token, added (concatenated) to the input again to generate the third token, and so on. GPT repeats the procedure until the special token *<EndOfSequence>* is generated or reaches the maximum context length.

At training time, first the model generates the whole sequence. Next, the training algorithm compares it with the sequence label in order to compute the loss function (typically, the cross-entropy loss). In our planning context, we generate a sequence of actions, and the label is a valid plan. The sequence can be derived using different generation strategies (Welleck et al. 2020):

- **Greedy Generation**: at each generation step, GPT outputs the token with the highest probability.

- **Multibeam N generation**: GPT keeps the most likely N sequences, and chooses the sequence that has the highest probability.

- **Top-P (nucleus) sampling**: at each step, GPT samples the most likely tokens whose cumulative probability reaches a given probability P.

## Architecture of PLANGPT

Our aim is to compute an effective general policy for a planning domain by training from scratch a custom GPT architecture for that domain. Given a PDDL problem with its initial and goal fluents as input, the objective is to generate a sequence of actions that solve the problem. We build a different model for each domain, training it from scratch using a set of training examples, each one made by an initial state, a set of goal fluents and the corresponding solution plan, which is the label of the example. In this section, we describe the preprocessing of the input data, the overall working of the model, how we train it, and how we evaluate the generated plans.

### Preprocessing and Tokeniser

In the preprocessing phase, we transform the initial state and the goal into a format suitable for GPT (i.e. a sequence of

tokens). At training time, we also include the tokenised version of the solution plan $P$ as the label of our training procedure. At inference time, the model has the objective of generating $P$ in the same format.

To achieve this, the tokeniser splits each input fluent in its components (the predicate name and its objects); then these tokens are concatenated to obtain a token sequence representing both the initial state and the goal fluents. Similarly, we apply this procedure for each action that is generated and provided as input in the incremental generation of a solution plan: the tokens of an action are its name and the objects of the action, and a plan is a sequence of such tokens. For example, for fluent (*At Truck1 Loc1*) we have three tokens: *At*, *Truck1* and *Loc1*; for action (*Drive Truck1 Loc1 Loc3*), four tokens: *Drive*, *Truck1*, *Loc1* and *Loc3*.

The tokens of the initial state, the goal fluents, and the (already generated) action sequence are then concatenated to obtain a single sequence for GPT using some special tokens as follows:

- **<start>** to mark the start of the initial state.
- **<goal>** to mark the end of the initial state and the beginning of the goal fluents.
- **<actions>** to indicate the end of the goal fluents and the beginning of the tokens of the action sequence.
- **<end>** to mark the end of the action sequence and, consequently, of the entire plan generated to solve the planning problem.

As in many GPT models, the tokenisation and preparation of the model input are performed by WordPiece (Devlin et al. 2019). Since GPT models have a predefined vocabulary, i.e., a predefined set of tokens, we defined a predefined set of objects for each domain during training and instantiated all the PDDL action schemas and predicates using those objects to obtain the PLANGPT vocabulary of the object and action tokens. Although the predefined vocabulary is sensitive to the object names, in our architecture this operation only requires setting a maximum number of objects. This is because we use a mapping algorithm to translate new names into predefined ones. The algorithm takes as input a PDDL problem and retrieves all the objects, then checks if these objects are in the GPT vocabulary. If an object is not in the vocabulary, it is substituted with an unused object of the same type in the vocabulary.

## PLANGPT Models

Figure 1 shows the architecture of our system, PLANGPT, in which we use the latest open-source version of GPT (GPT-2).[1] Given in input the initial state and goal of a planning problem in a planning domain, PLANGPT generates a sequence of ground actions (each one tokenised as described above) forming a plan to reach the goal from the initial state. First, the input is tokenised as described in the previous section. After tokenisation, the embedding layer converts the tokens into embedding vectors, the decoder stack analyses

---

[1] GPT-2 is significantly smaller than recent GPT versions, and hence much less demanding in terms of training data and required computational resources for training.

the input sequence, and the final layer outputs the first token of the plan. Then, PLANGPT adds the generated token to the input sequence and repeats the whole process for generating the second token, and so on. Each token generated by GPT is the name of an action or one of its objects. For example, if the output of PLANGPT is the sequence *Drive, Truck1, Loc1, Loc3, ..., <end>*, the first action of the generated plan is *(Drive Truck1 Loc1 Loc3)*. The tokens of each output action are generated one after the other. E.g., first PLANGPT generates *Drive*, then it adds *Drive* to the input sequence and outputs *Truck1*, and so on. This is repeated until the end-of-sequence token *<end>* is generated.

Tipically, GPT models are trained to generate all the tokens starting from token *<start>*, i.e., to replicate the entire input sequence, which in our case represents the initial state, the goal, and a solution plan. Since our goal is to generate only a plan, we are not interested in learning how to replicate the tokens of the initial state and the goal. Therefore, we include a loss masking mechanism in the training procedure to prevent the model from learning to generate these tokens.

Each training example consists of the sequence of tokens derived from a planning problem (initial state and goal) and its label is the sequence of tokens from a solution plan solving the problem. At training time, the model generates a sequence of tokens corresponding to a sequence of actions. Such a sequence is compared against the example label to compute the loss function and adjust the network weights through backpropagation. As loss function we adopt the widely used cross-entropy.

## Planning Coverage for Early Stopping in Training

Generally speaking, the use of the cross-entropy (CE) forces the model to mimic the example label. Each time the model generates a token, this token is compared with the corresponding one belonging to the label. From this comparison the loss is calculated, and then, the backpropagation algorithm modifies the weights in order to generate a sequence of tokens that is the closest possible to the label.

However, this process is not fully adequate for learning to solve planning tasks because a planning problem can be solved by different plans. With the cross-entropy loss function, we may observe an error in the generated (tokenised) plan just because it is not totally identical to the one that was used as label. This problem is exacerbated by the tendency of deep learning models to overfit the training data. With an overfitted model, we may have a model that is capable of generating plans for the training problems, but is uncapable of solving other similar problems in the test set.

A typical technique to prevent overfitting is the *Early Stopping*. The mechanism uses a validation set of examples that are not used for training. If the loss value for the validation set (the validation loss) increases, which is a typical evidence of overfitting, the model continues to train for a fixed number of epochs. After these epochs, if the validation loss has not improved, the model restores the weights that obtained the best performance on the validation set and the training stops.

Since, for planning, optimising the standard cross-entropy on the validation set suffers the problem outlined above, we

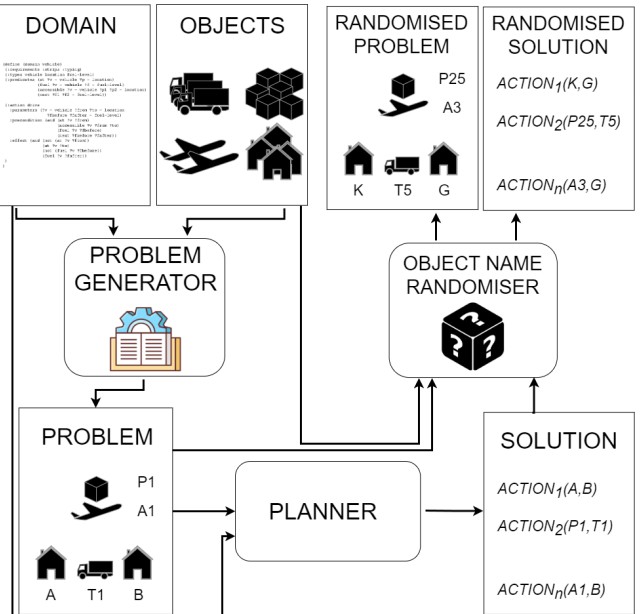

Figure 2: Dataset generation procedure. Given a PDDL domain and a set of objects, the problem generator outputs a PDDL problem in that domain; then the planner generates up to four solution plans solving it; finally the objects names of the problem and of the corresponding plans are randomised.

designed a new early stopping technique. Our technique, called **Planning Coverage Early Stopping** (CES for short), evaluates the capability of *solving the planning problems* in a validation set with the current learned model. This metric is based on verifying the correctness of the plans generated by the model for the planning problems in the validation set. The verification is performed using the PDDL specification of the actions (preconditions and effects) through the standard validator VAL (Howey, Long, and Fox 2004).

At the end of each training epoch, the model generates the solution plans for the validation problems. If at least one action in a plan is not applicable or at least one goal fluents is not reached, the plan is considered incorrect, otherwise it is valid. The coverage metric value is defined as the percentage of valid plans over the total number of generated plans.

The use of CES helps the model to generate valid plans rather than plans identical to ones labelling the training set. The CES metric is evaluated at each training epoch; if the CES value has not improved for a predefined number of epochs, the training stops and we select the model's weights which obtained the best performance in terms of the coverage metric.

## Dataset Generation

In this section, we describe the procedure depicted in Figure 2 to build the dataset used for training our GPT model. First, we generate $70,000$ planning problems (written in PDDL) for the considered domain using a problem generator; we used the one proposed in (Seipp, Torralba, and Hoff-

mann 2022). Depending on the number of objects involved, we have problems of different difficulty. We chose the number of objects following the setups of the International Planning Competition (IPC).[2] However, with such setups and the problem generators we could derive a class of problems that is too specific for training, limiting the generalisation capability of the learned model. When we observed this issue for the domain considered, we generated additional problems through a revised problem generator. For instance, the generator available for LOGISTICS always creates problems in which all packages of the problem must be transferred to a location specified in the goal. To address this bias, we also create scenarios where the goal does not encompass all the packages specified in the planning problem.

For a typed domain, we incorporated the type predicates into the initial states of the problems. E.g., for problems in LOGISTICS, predicates such as *(City City1)* or *(Package Package1)* are added to the initial state. This helps the model to associate each object name with its corresponding type.

For each generated problem, we compute different suboptimal solutions (four in our implementation). In this way, we show to the system that a single problem can have more than one valid plan. Furthermore, generating multiple solutions for a single problem augment the number of training samples. To obtain multiple plans, we used LPG (Gerevini and Serina 2002), but other planners could be used (Richter and Westphal 2010; Lipovetzky and Geffner 2017; Helmert 2006).

Then, we randomise the object names to mitigate potential biases in the generated problems and plans. The names randomisation is performed by replacing each object name *obj* with a name of an object of the same type of *obj* randomly taken from the vocabulary. This step is important because it prevents a deep learning model to learn biases tied to the conventions used in problem generators. In LOGISTICS this issue arises for the following reason: the generator names trucks and cities with increasing numbers and assigns them in increasing order. E.g., assume we consider problems with three cities (*City1, City2, City3*) and one truck for each city; the generator always set in the initial states *Truck1* at *City1*, *Truck2* at *City2* and *Truck3* at *City3*. Training a model using only problems following this convention limits its generalisation capability, since it would provide wrong results for instances following other conventions.

In addition to LOGISTICS, we built datasets for other seven well-known benchmark domains (briefly described in the supplementary material): BLOCKSWORLD, DEPOTS, DRIVERLOG, FLOORTILE, SATELLITE, VISITALL and ZENOTRAVEL.

We also analysed the previous biases in the IPC benchmarks and available generators of these domains. We observed that all the IPC problems of BLOCKSWORLD have only one tower to build, while BLOCKSWORLD generator always creates problems with more than one tower to build. Therefore, we used a variant of the standard generator where every problem requires to build from one to five towers.

In ZENOTRAVEL, planes consume fuel transporting peo-

---

[2]https://www.icaps-conference.org/competitions/.

ple, and in general, a varying fuel level (among those available) is assigned to each plane in the initial state of a problem. However, in the ZENOTRAVEL problem generator, every plane has zero fuel in the initial state. A model trained with such problems could learn a simplified version of the domain in which all planes used must always be refuelled, without an understanding of the overall fuel management. To solve this bias, we extended the ZENOTRAVEL generator to randomise the initial fuel level of each aircraft.

For VISITALL, as in the approach of (Ståhlberg, Bonet, and Geffner 2022b), we used the IPC-2011 optimal track problems, and we generated problems with rectangular grids of different sizes and different percentages of tiles to visit.

## Experimental Results

We trained a custom GPT model for each of the eight domains indicated above using GPT-2 Small, which has 12 blocks with 12 heads each, for a total of about 83M parameters. The training hyperparameters are available in the Supplementary Material. We also tested bigger GPT configurations (which require a higher number of training instances, more training time, and more computational power) without obtaining significantly better results. Our models are trained on a NVIDIA A100 GPU of 40 GB.[3] We tested the three standard generation strategies previously described: Greedy, Multibeam N generation (setting N = 10) and Sampling Top-P (setting P = 0.9 and generating 10 sequences).

In the following, we evaluate our GPT-based models in terms of percentage of valid generated plans (*coverage*). The generation takes, on average, less than 3 seconds, with a maximum of 20 seconds for FLOORTILE using Sampling. We also experimentally compare our approach with Plansformer (Pallagani et al. 2023), the best performing transformer-based model applied to automated planning, and with a state-of-the-art approach for learning general policies based on GNN (Ståhlberg, Bonet, and Geffner 2022b). This comparison is perfomed in terms of correct plans, and scores *IPCScore-Quality* and *IPCScore-Agile*, as defined in the last planning competition (IPC 2023) and reported in the supplementary material.

### Effectiveness in Valid Plan Generation

Table 1 shows the results of PLANGPT with and without the Planning Coverage Early Stopping (CES) for the Greedy, Multibeam and Sampling generation strategies. For this evaluation, we used a test set of more than 6000 problems for each considered domain; this test set, indicated with Tset, was created using the available generators modified as described above to avoid original biases previously discussed.

Our system obtains very good results for most of the domains considered. In particular, with Sampling the coverage is higher than 90% for every domain except LOGISTICS, where we have 77.3% of coverage. PLANGPT solves all the BLOCKSWORLD and VISITALL problems and 99.6% of the FLOORTILE problems with CES. With Multibeam and

---

[3]The code, the datasets and the models will be made available after acceptance of this paper.

| Domain | Greedy CE | Greedy CES | Multibeam CE | Multibeam CES | Sampling CE | Sampling CES |
|---|---|---|---|---|---|---|
| BLOCKSWORLD | 98.8 | 99.5 | 99.4 | 99.6 | **100.0** | **100.0** |
| DEPOTS | 72.9 | 78.7 | 77.1 | 85.4 | 90.3 | **94.5** |
| DRIVERLOG | 61.3 | 68.4 | 73.0 | 80.8 | 94.7 | **96.5** |
| FLOORTILE | 92.9 | 94.4 | 96.9 | 96.6 | 98.2 | **99.6** |
| LOGISTICS | 63.3 | 66.1 | 62.8 | 63.7 | 76.3 | **77.3** |
| SATELLITE | 68.0 | 75.3 | 71.6 | 78.3 | 81.3 | **90.1** |
| VISITALL | 94.0 | 94.0 | 97.8 | 97.8 | 99.9 | **100.0** |
| ZENOTRAVEL | 82.7 | 82.7 | 87.3 | 87.3 | **94.7** | **94.7** |

Table 1: Coverage for each domain with the greedy, multibeam and sampling generation of PLANGPT using standard Cross Entropy without (CE column) and with the Coverage Early Stopping (CES column) using the Tset test set.

Greedy we have a lower performance, but the coverage percentage is never lower than 60%.

We now evaluate the effectiveness of our coverage early stopping (CES) technique, analyzing the coverage performance with and without its utilization on the Tset test set. The results are presented in Table 1. The use of CES improves performance in all domains except ZENOTRAVEL, where the performance remains the same. In particular, we have a remarkable improvement for SATELLITE with all three generation strategies (7.3 points with greedy, 6.7 with multibeam, and 8.8 with sampling), DEPOTS and DRIVERLOG. Even for domains where PLANGPT obtains very high performance without CES, such as FLOORTILE, VISITALL and BLOCKSWORLD, with CES we still have a small improvement. These results confirm the usefulness of including our planning evaluation technique in the training process.

In Figure 3 we examine the behaviour of the cross-entropy loss function and the use of CES for three domains during training (we report only these three domains for the sake of clarity; the results for the other domains are available in the Supplementary Material). The black cross on the curves indicates when the training stops using the standard cross-entropy loss evaluated on a validation set (1000 randomly generated problems for each domain not used for training PLANGPT) as the early stopping metric. The red star markers indicates when the model stops the training using CES. For all three domains, using CES leads to train for a higher number of epochs w.r.t. not using it (i.e., with the standard cross-entropy technique). In these additional epochs, the loss function value worsens. Despite this worsening of the loss function, the coverage increases until the number of epochs indicated by the red star marker is reached. This shows that using the CES improves the training process, obtaining higher coverage.

The experimental results of Table 1 and Figure 3 indicate that the standard loss function of GPT-2 is not fully adequate to learn planning policies. In fact, the experimental results show that an improvement in the plan generation can be obtained with a worsening of the loss function at training time. As we already noticed, a possible reason is that the standard loss forces the model to imitate the target plan (the sample label), limiting the model capabilities of generating a valid

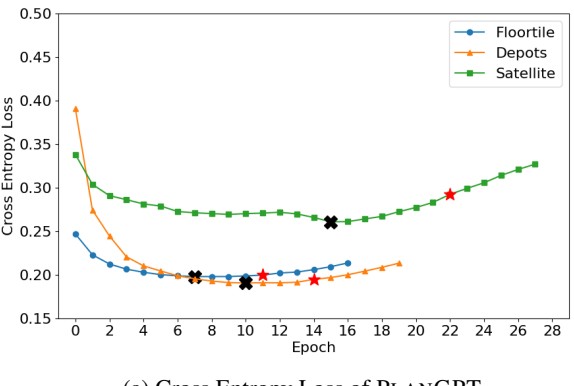

(a) Cross Entropy Loss of PLANGPT.

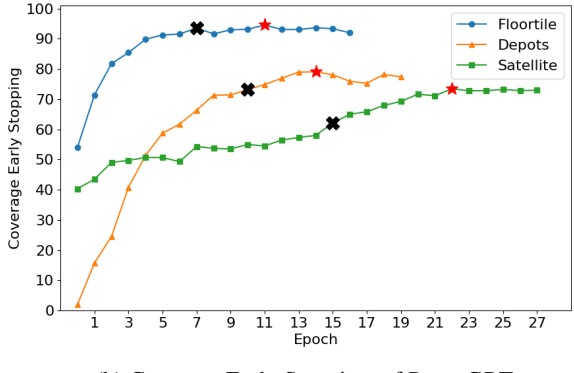

(b) Coverage Early Stopping  of PLANGPT.

Figure 3: Cross Entropy Loss (on the left) and Coverage Early Stopping  (on the right) for each epochs in the training phase of PLANGPT for DEPOTS, FLOORTILE, and SATELLITE domain on the validation set. The black marker indicates the training termination when the Cross Entropy Loss is at its minimum. Instead, the red marker indicates the termination of the training when Coverage Early Stopping  is at its minimum.

plan that is different from the target one.

We have also performed an analysis of the invalid plans generated by PLANGPT with aim of understanding its main mistakes. Most of the errors in the invalid plans are related to a violation of a precondition. In particular, for LOGISTICS the trained GPT-based model selects an object that is not in the correct location for the truck/plane loading action. Therefore, we argue that the main difficulty for the model is understanding the relation of the objects involved in a single action (the truck and the package must be in the same position to perform a *load-truck* action). For the invalid plans in SATELLITE, PLANGPT generates *take-image* actions that have preconditions of type *supports instrument mode* unsatisfied. This is because before these actions PLANGPT powers on, points, and calibrates the wrong instruments (that do not support the mode needed later by the *take-image* actions). For the invalid plans in DRIVERLOG, PLANGPT generates *walk* actions moving a driver between locations that are not connected (as requested by the action preconditions).

Finally, we have evaluated PLANGPT when used to perform *plan completion* tasks rather than plan generation from scratch. In this setting, in input we have the additional information of a plan prefix (an initial sub-sequence of its actions), and we ask the system to complete the plan. Overall the results are promising, reaching performances higher than when planning from scratch. In particular, in LOGISTICS (the domain with the worst performance in Table 1), using the Sampling strategy and an input plan prefix of 10%, 20%, 40% and 60% of a valid plan, the obtained performance in term of coverage is 79.9%, 83.1%, 86.1%, and 90%, respectively.

### Evaluation with the IPC Benchmarks

In this section, we evaluate PLANGPT using benchmark problems from the International Planning Competition (IPC). For this experiment we use CES and the Sampling generation strategy, which we observed to perform generally better than the other two implemented strategies.

When an object name in an IPC problem is not in the PLANGPT vocabulary, a name for that object is randomly selected from the vocabulary, saving it into a conversion table. However, if the number of objects is higher than those in the vocabulary, the problem cannot be attempted by PLANGPT. This is the case only for 4% of the IPC problems in the eight considered domains (2 problems in DEPOTS, 3 in SATELLITE and 2 in ZENOTRAVEL). In the following, we will consider both sets of test sets, the original set without the problems that PLANGPT cannot attempt (IPC$^-$ test set) and the original one (IPC test set).

The results of this experiment are in Table 2. PLANGPT solves all IPC problems in the domains of BLOCKSWORLD, FLOORTILE and ZENOTRAVEL, and a very high percentage of problems in the domains of DEPOTS, VISITALL and DRIVERLOG. These remarkable results are especially interesting for FLOORTILE because its IPC problems have numerous dead-ends and different grid conformations, which make them hard to solve for state-of-the art planners such as LAMA (Richter and Westphal 2010) and FastDownward (Helmert 2006) (e.g., LAMA solves only 2 of the 20 FLOORTILE problems with run time limit of 10 minutes). The domain for which we observe the lowest performance is LOGISTICS, where only 53.3% of IPC problems are correctly solved. The observed performances for the original and the restricted (IPC$^-$) test sets are similar, with lower performance for the original set because the 4% problems that are not attempted are counted as unsolved problems in the results for the original test set.

### Comparison with the State of the Art

In this section, we compare PLANGPT and state-of-the-art deep learning models for computing general policies. We consider Plansformer (Pallagani et al. 2023) and the Graph Neural Networks proposed in (Ståhlberg, Bonet, and Geffner 2022b).

| Domain | IPC$^-$ test set | IPC test set |
|---|---|---|
| BLOCKSWORLD | 100.0 | 100.0 |
| DEPOTS | 95.0 | 86.4 |
| DRIVERLOG | 95.0 | 95.0 |
| FLOORTILE | 100.0 | 100.0 |
| LOGISTICS | 53.3 | 53.3 |
| SATELLITE | 70.6 | 60.0 |
| VISITALL | 95.0 | 95.0 |
| ZENOTRAVEL | 100.0 | 90.0 |

Table 2: Coverage of PLANGPT using the Sampling strategy and CES on the IPC/IPC$^-$ test sets.

| Domain | Coverage | | IPC-A | | IPC-Q | |
|---|---|---|---|---|---|---|
| | GPT | GNN | GPT | GNN | GPT | GNN |
| BLOCKS | **100.0** | 26.2 | **6292.5** | 1247.4 | **6597.1** | 1611.0 |
| LOGISTICS | **77.3** | 21.6 | **4752.2** | 791.7 | **5125.1** | 772.1 |
| VISITALL | **100.0** | 96.0 | **5754.5** | 3176.4 | **6046.4** | 6002.0 |

Table 3: Comparison of PLANGPT (GPT) and GNN in terms of problem coverage, *IPCScore-Agile* (IPC-A) and *IPCScore-Quality* (IPC-Q) on the `Tset` test set. BLOCKSWORLD is abbreviated with BLOCKS.

Plansformer is a transformer trained on code written in several programming languages (CodeT5) and fine-tuned on planning problems. In general we observed that our PLANGPT models perform much better than the available models of Plansformer. For instance, on the IPC problems of BLOCKSWORLD and DRIVERLOG the coverage results are 100% versus 11%, and 90% versus 5%, respectively. Plansformer's inability to generalise to complex instances (the IPC benchmarks) could be explained by the excessive simplicity of the problems in its training set (up to 5 blocks in BLOCKSWORLD compared to 20 in our training dataset, and up to 4 packages in DRIVERLOG compared to 25 in our training). We tried to re-build Plansformer by fine-tuning CodeT5 using our LOGISTICS and DRIVERLOG datasets. Even in this case, Plansformer obtained much lower performance for the two tested domains (coverage 30% and 5% versus 53.3% and 96.5% of PLANGPT).

We now compare our GPT-based approach and the approach based on Graph Neural Networks (GNNs) proposed in (Ståhlberg, Bonet, and Geffner 2022b), which in the following is indicated simply with GNN. For this comparison we use three domains: BLOCKSWORLD, LOGISTICS and VISITALL.[4]

Starting from the problem initial state, GNN evaluates the successor states using a Graph Neural Network as heuristic function, and chooses the action that leads to the best successor state; this is repeated for such successor state, and so on

---

[4]We could not use the other domains examined in (Ståhlberg, Bonet, and Geffner 2022b) because either they are too simple, or no generator is available, or they have particularly long lists of predicates in the problems that exceed the PLANGPT context window (2048 tokens). This implementation limitation could be solved by using GPT models with larger context windows.

until a state satisfying the goal is reached. The GNN models are trained with the IPC problems, augmenting the training set with traces obtained during the heuristic search of the planner BFWS (Lipovetzky and Geffner 2017). Therefore, we can not use the IPC problems as *test* set, and so we use our test set (`Tset`) as benchmark.

Table 3 shows the performance of PLANGPT and GNN in terms of coverage and IPC scores. GNN solves only 26.40% of the BLOCKSWORLD instances while PLANGPT solves all of them. By analysing the generated plans, we notice that GNN is unable to solve many problems where the agent has to build more than one tower. Given that the GNN models were trained with the IPC benchmarks where the goal of every problem has only one tower of blocks, it appears that GNN is unable to generalise because it learnt this bias.

For LOGISTICS, PLANGPT obtains a coverage of 77% versus 21.6% of GNN. The authors of GNN notice that LOGISTICS is a challenging domain for GNN due to its belonging to the C3 logic fragment (Ståhlberg, Bonet, and Geffner 2022b). For this reason, they also modified this domain (changing the used fragment of logic to C2), adding a predicate to link packages, trucks, and planes to locations in the problems. With this modification of the domain, coverage increases to 44.7%, which is still lower than the coverage result of PLANGPT. We also trained PLANGPT with this modified version of LOGISTICS, observing a coverage performance similar to the one of GNN.

For VISITALL, PLANGPT obtains a coverage of 100% versus 96% of GNN.

Regarding the comparison in terms of IPC scores reported in Table 3, we observe that, for the considered domains, PLANGPT performs generally better than GNN in terms of both run time to generate a valid plan (IPC-A column) and length of the generated plan (IPC-Q column). Note that the definitions of the IPC scores take account of the problems that are unsolved.

## Conclusions

We have investigated generalised planning as a deep learning task using transformer-based architectures. We propose a system based on GPT, called PLANGPT, that learns to solve an extensive class of problems for a given planning domain. Our training procedure exploits a technique that we designed to take into account the planning capability of the model in the validation phase, which we show helps to increase the performance of the trained system w.r.t. just using the standard cross-entropy loss.

An experimental analysis demonstrates the effectiveness of our approach. For several domains, PLANGPT solves the large majority of the IPC benchmark problems, as well of other larger test sets, and it achieves better or comparable performance w.r.t. state-of-the-art approaches based on Transformers or Graph Neural Networks.

Current and future work includes improving the training process through a tighter integration of planning knowledge in the loss function, and to overcome the current limits due to maximum number of objects in the vocabulary and the length of the context window.

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
