# OpenReview forum: "Learning General Policies for Planning through GPT Models"
_icaps-conference.org/ICAPS/2024/Conference — ICAPS 2024_

### Official Review · Reviewer_ufRi · 2024-01-05

**Significance And Importance:** 2
**Soundness:** 2
**Novelty:** 2
**Clarity:** 2
**Confidence:** 4

**Weaknesses:**

-1: Major weaknesses requiring significant work to be addressed for the paper to be accepted.

**Contributions Of The Paper:**

The paper uses GPT-2 to learn generalised policies for a domain from plan examples for training instances in this domain. It develops an approach where GPT-2 learns to predict the next action in a plan from a context consisting in the initial/goal state and previous actions in the plan. To avoid overfitting, a stopping criterion is developed which evaluates the validity of generated plans on a validation set using the plan validator VAL. Results obtained by training on random instances for IPC domains and plan examples generated by LPG are encouraging and better than other LLM approaches this reviewer is aware of.

**Ethical Considerations:**

(1) Not Applicable: The paper does not have any ethical considerations to address

**Nomination For Best Paper:**

No

**Overall Evaluation:**

-1: (weak reject)

**Questions For Authors:**

a. When you refer to the IPC domains/instances, which track and which year are you referring to exactly? And if not the 2023 learning track, why not?

b. Why didn't you train and evaluate Muninn and PlanGPT on the same training and test instances?

c. Is planGPT able to solve larger instances than those it was trained on, and if it is, why is there no experimental results evaluating this?

d. Can you please describe the performance of PlanGPT wrt the teacher planner?

**Reproducibility:**

4: Authors promise to release code and domains (whichever apply).

**Strengths Of The Paper:**

Given the success of LLMs elsewhere in AI, it is important for the planning community to understand whether and how these might be used in planning. The paper contributes to this.

The approach makes sense: learn the language of plans in a domain in a supervised manner, from plans generated by existing planners and cross-entropy loss to mimic those plans.

Moreover the approach seems to be far more successful than other LLM approaches to planning (and in the case of floortile, the results are surprisingly good).

**Weaknesses Of The Paper:**

There are important weaknesses, having to do with lack of rigour in the presentation of the approach and its evaluation.

The first weakness is that the paper is quite informal overall. I am not sure I would be able to reproduce the approach from the explanations given in the text. For instance, not even the loss function is formally given. As another example, the preferred Top-P generation strategy is described as follows: "at each step, GPT samples the most likely tokens whose cumulative probability reaches a given probability P". What does it do then with these tokens? This is clear with other two strategies, but not here.  I am just giving examples here, but almost nothing is formally described.

The background subsection on GPT2 in terms of matrix operations suffers from the opposite issue: it isn't sufficiently insightful. Given that LLMs are still relatively unexplored in planning, it would be important that the background provides sufficient insight to the average planning reader into how GPT works and how this relates to planning. Yet, I do not think anyone who doesn't already know GPT will be able to derive much understanding from the current GPT subsection.

The second main weakness is that there are some important issues with the experiments, including missing information (even after having read the supplementary material), and comparisons that do not seem fair. As a result it is difficult to evaluate the performance of this approach in comparison with the state of the art whether in model-based or data-driven planning.

1. The number of objects in the training instances are said to have been chosen according to the settings of the IPC. But which IPC track/year are we talking about? It seems it's not the 2023 learning track, as one might have expected from current work on learning for planning. The size of the training problems is given in appendix. However, the size of the test problems isn't. Is that the same size as for the training problems? These aren't very large in comparison with problems solved by systems such as Muninn or Goose. Given the statement that "Once PlanGPT has been trained, it can be used to generate a solution plan for an input problem instance in that domain" I would have expected some experiments with problems of larger size to see whether the learnt policies generalises to larger problems. If it doesn't then the main gain is faster speed than existing planners on the domains and sizes trained on.

2. I would have expected to see a comparison (coverage, speed, quality) with the planner used for training on problems of same and larger sizes than those trained on.

3. The comparison with Muninn presented in the paper seems unfair. As far as I understand from the paragraph around line 690, you are training Muninn with the IPC problems -- so it's a different training set than the one used to training PlanGPT -- but for reasons which I do not understand you say that you cannot use the IPC problems as evaluation problems -- and so instead, you test Muninn on the problems from the PlanGPT test set (which, if I am correct in 1. are from the same distribution as the PlanGPT training set). It would have been fair to have the same training and test sets (or at least distributions) for both approaches. As you observed with Blocksworld, it is therefore not surprising that Munnin performs worse. To me, this comparison is invalid. The comparison in the supplementary material where both approaches are in fact evaluated on IPC problem paints a totally different picture -- but still not ideal in part because the training set given to PlanGPT is still much more comprehensive. Moreover the reasons given for not performing a more comprehensive comparison including other domains isn't really convincing. Muninn competed in the 2023 learning track so you could have used those 10 domains for comparison. All of them have random generators.

Even though I would not count this as a weakness in this case, the contributions are only moderately novel. The approach makes sense, but it is not too distant from works that have used supervised or imitation learning for planning, except that the network is different.


Other less important issues:

The motivation in the introduction is weak. The justification given for the work is that most learning for planning approaches learn heuristics instead of policies and that many of them do so from images or have restrictions. But in fact, many approaches that learn heuristics could easily be adapted to learn policies instead, and do not have the mentioned restrictions. I think the motivation should be the need to understand where LLMs best fit in the planning landscape, and which planning artefacts they're best placed to learn.


The approach seems to require far more data than other approaches for learning policies or heuristics (70,000 training problems, 83M parameters).


Details:

The introduction is praising Plansformer, stating it obtained promising results. Judging from the description of the dataset on page 4 of their paper, the size of problems is much too small for the results to qualify as promising.

The word fluent is used without being introduced. These days, and in particular in PDDL, fluent is typically used to refer to numeric variables. Here I think you just mean proposition.

The legend of Figure 1 mentions I_0 and I_1. I_i is used in Section 2 to denote vectors. I would use different notations to avoid confusion.

You say that all the IPC problems of Blocksworld have 1 tower. What track are you referring to? The Blocksworld generator definitely supports multiple towers.

Seipp et al reference: Please provide the link to the repo.

---

> ### Author Rebuttal · Authors · 2024-01-27
>
> We thank the reviewer for the helpful comments.
>
> We will improve the presentation as suggested. The loss function for a generated token is the standard cross-entropy, as indicated in the paper; for the full plan, it is the average over all plan tokens. We will formalize this.  Please see also the response to Q1 of 1st Rev.
>
> Qa. We selected domains and configurations from IPCs that are used also in other related papers to facilitate the comparison. As in Stahlberg-et-al we include Blocksworld (IPC00), Logistics (IPC00) and VisitAll (IPC14-opt). Zenotravel (IPC02-aut) is used for comparison with Pallagani-et-al. We also include Depots and Driverlog (IPC02-aut), Satellite (IPC04-plain) and Floortile (IPC14-sat). The last 2 domains are in the IPC23 learning track. The papers about the compared systems don’t use more domains than ours.
>
> We did not enter IPC23 (results analysis published Sept 20th) and we became aware of the problem generators for the IPC23 track too late for the ICAPS24 deadline.
>
> Qb. Muninn is a planner performing search guided by a general policy (the GNNs by Stahlberg et al) while PlanGPT is a general policy without search. We can compare only the policies.
>
> GPT requires more training data than a GNN. We did not train the GNNs for computing general policy (we will revise a misleading sentence in the paper): we used the GNN models provided to us by the authors, who trained/optimized the GNNs using the same distributions as for PlanGPT (except one commented below). Training the GNNs with the large datasets of PlanGPT could lead to problems like overfitting.
>
> Testing the GNNs using the IPC instances (IPCset) is methodologically questionable because the GNNs were *trained* by the authors with instances that include 50% of the IPCset. However, as shown in Table 4 (Sup Material), PlanGPT obtains better or comparable results even for IPCset. We could move them in the paper.
>
> We agree that the comparison for Blocksworld may be unfair since in this case the distribution of the test instances is not the same in the two systems. We have repeated the testing after removing from Tset all instances not in the IPC00 distribution (as in Stahlberg et al). The results are still in favour of PlanGPT: coverage 100% vs 81%; IPCQ 1847 vs 1611; IPCA 1763 vs 1093. We will revise the paper.
> For Logistics and VisitalAll, the Tset instances have the same distribution of IPCset.
>
> Qc-d. Please see responses to Q1 of 1st Rev and to “Comparison with planners” of 2nd Rev.

---

### Official Review · Reviewer_kCZo · 2024-01-22

**Significance And Importance:** 3
**Soundness:** 3
**Novelty:** 3
**Clarity:** 4
**Overall Evaluation:** 2
**Confidence:** 4

**Weaknesses:**

2: No major or minor weaknesses.

**Contributions Of The Paper:**

The authors investigate classical planning as a deep learning task using transformers. They propose PlanGPT, a planner that learns to solve problem instances for a given planning domain. They propose a novel early stopping method and perform a variety of experiments.

**Ethical Considerations:**

(1) Not Applicable: The paper does not have any ethical considerations to address

**Nomination For Best Paper:**

No

**Questions For Authors:**

- Is there a reason the authors did not try to create an even more general planner by including the domain in the training process as well? Training GPT models for each domain seems misaligned with what the AI planning community has worked towards, which is general heuristics that can be used across domains.
- On pg 6 it is mentioned that "bigger GPT configurations" were tried with limited results. Could you be more specific about what tests were performed, and on what size architectures?
- Could the method be extended to optimal planning (generating optimal plans (i.e., those with minimal actions)) by simply training on optimal solutions to planning problems?

**Reproducibility:**

3: Authors describe the implementation and domains in sufficient detail.

**Strengths Of The Paper:**

The paper is well-written and clearly organized. It is very relevant to the ICAPS community, and the topic is interesting.

**Weaknesses Of The Paper:**

- The paper appears to be technically correct, though many of the low-level technical details do not appear in the text.
- It is not clear how the proposed method performs against state-of-the-art classical planners (FF, etc) in terms of runtime and number of instances solved. The reviewer would welcome this comparison as well.

---

> ### Author Rebuttal · Authors · 2024-01-27
>
> We thank the reviewer for the helpful comments.
>
> Q1. We agree that having a general policy that works well on different domains is an interesting research goal to pursue; investigating this is part of our future work. However, in the literature the generation of general policies is usually investigated for single domains (Stahlberg-et-al, Toyer-et-al, Groshev-et-al).  To our knowledge, our approach is the first that trains a GPT model from scratch. Therefore, we first wanted to investigate if this approach works on single domains.  Furthermore, a multi-domain model, as pointed out by the reviewer, needs to include the domain in a format that is suitable to process it for GPT. This would require a much larger context window, which would significantly increase the complexity of the training process.
>
> Q2. We considered GPT-2 medium (345M parameters) on Logistics and CodeT5 (770M parameters) on Driverlog, using the same data and experimental setup we used for PlanGPT. On Tset, GPT-2 medium obtained a coverage of only 5% (PlanGPT 77%), CodeT5 obtained a coverage of 20% (PlanGPT  96%).
>
> Q3. This is an interesting idea that we are currently exploring. We believe that training GPT on optimal plans can lead to significant improvements in the quality of the generated plans. However, PlanGPT does not guarantee plan optimality, and generating a dataset of optimal plans, in general, is computationally more expensive.
>
> Comparison with Planners:
>
> A fair performance comparison is difficult since PlanGPT is a general policy, not a planner. Moreover, it is written in Python and works on a GPU while SOTA planners are often written in C/C++ and work on CPUs.
> In any case, we have compared the performance of PlanGPT with respect to LPG (the planner we used for generating our training instances) and other SOTA planners, with a time limit of 5 minutes. The summary results over all the considered domains are:
>
> ||Coverage(187)|IPCQ|IPCA|
> |-|-|-|-|
> |planGPT|158|154.3|132.8|
> |LAMA|168|135.2|165.3|
> |FD|165|129.2|160.1|
> |LPG|185|119.2|182.6|
> |BFWS|183|22.6|170.4|
>
> Regarding PlanGPT and the teacher planner LPG, they have similar coverage for all domains except for Logistics (16 problems solved by PlanGPT against 30 solved by the planners) and Satellite (12 problems solved by PlanGPT against 20 solved by the planners).
> We note that PlanGPT successfully solves all 20 problems of Floortile, LPG solves 18, and LAMA/FD only 2. In terms of IPCQ, PlanGPT performs better than all planners.

---

### Official Review · Reviewer_4tAp · 2024-01-22

**Significance And Importance:** 3
**Soundness:** 3
**Novelty:** 3
**Clarity:** 3
**Overall Evaluation:** 2
**Confidence:** 5

**Weaknesses:**

1: Minor weaknesses that are easily fixable.

**Contributions Of The Paper:**

This is an important paper. While there have been many works showing that ChatGPT and the like don't reason or plan too well, to say the least, the authors show that the underlying architecture, basically transformers, can produce meaningful, non-trivial plans over many existing PDDL domains, once the transformers are training in a supervised way with domain instances and plans. In this way, through the use of a decoder-architecture, the transformer can output new ground plans, from new domain instances.

I like the work in particular because in my team we have also been exploring the learning/computation of general plans with transformers but never managed to get results that are as good (so this is not published). Of course, some of the design choices and the training were not the same, although, as a plus of the paper, the design choices made are all pretty direct: the way states and goals are encoded, and object ocurrences are embedded, etc.

**Ethical Considerations:**

(1) Not Applicable: The paper does not have any ethical considerations to address

**Nomination For Best Paper:**

No

**Questions For Authors:**

1. Please answer briefly A,B,C above in your rebuttal too .
2. Did you try learning a value function V instead of a ground plan for each instance? Any obvious design choices that did not work?
3. Do you use multiple plans for each problem during training?
4. What are you getting actually from the tokenizer?
5. In the sampling strategy is there any pruning done of wrong plans, or the plan that is sampled according to the method is the one that is considered for evaluation?

===

Post-rebuttal: thanks for the answers to my questions. I don't fully understand them all, but they help, and in any case, I think that the results achieved are surprisingly good and novel (in relation to transformer architectures), so I'm very positive about the work.

**Reproducibility:**

4: Authors promise to release code and domains (whichever apply).

**Strengths Of The Paper:**

Shows that one can do generalized planning with a transformer architecture by training on each domain separately. The results are not coverages of 100% in most of the domains, but they are pretty good and comparable to Stahlberg et al. So definitively, the architecture is learning something meaningful and can produce meaningful, and potentially long plans.

**Weaknesses Of The Paper:**

Some key choices in the experimental set up are not explained clearly and explicitly. E.g.,

A - Greedy generation, multibean N generation, Top-P sampling: are explained too briefly and too informally. All the details to reproduce these plan generation strategies should be spelled out in full technical detail. What are the values of the N, P parameters?

B - Distinction between training, validation, and testing data: you describe the pool of problems but not how this set of instances per domain is split into training, validation, and testing sets. This is critical.

C- It's not clear if like in Stahlberg et al, you are generalized to larger instances, or you are just generalizing to other instances of the same size as those used in training. Also, the comparison is informative, but it should be said probably that the amount of data and the training data is very different (70k plans for each domain vs. instances with up to 6-8 blocks, seems like a lot; although OK if needed for method to work, but worth saying).

---

> ### Author Rebuttal · Authors · 2024-01-27
>
> We thank the reviewer for the helpful comments.
>
> Q1. A.In the camera-ready version, if accepted, we will explain more in detail and formally all the generation strategies of GPT  (that are described in the cited paper). Regarding Top-P, the best strategy in PlanGPT, at each generation step, instead of choosing the most probable token in the vocabulary (Greedy strategy), GPT selects the smallest set of tokens whose cumulative probability exceeds the threshold P. Their probability is then refined considering only the selected tokens, and one is sampled using such probabilities. This is executed on N runs in parallel.
> As reported in the section of the experiments, we used P=0.9 and N=10.
>
> B.For each domain, we employed 63.000 problems for training, 1000 for validation and 7000 for the testing (Tset). We will include this information.
>
> C.We carried out preliminary experiments to test PlanGPT with problems involving higher numbers of objects in domain Zenotravel. The results show that PlanGPT struggles to generalize to larger instances. We conjecture that better generalization can be obtained with a more enhanced loss function; this is for future work.
>
> Q2. We have not investigated computing a value function over states through GPT. We believe employing this approach would slow down the plan generation process (as in Stahlberg et al), given the need to assess numerous states. Moreover, GPT is more apt to generative tasks.
>
> Q3. Yes. We generate up to 4 plans per problem (LPG generates multiple plans with increasing quality)  in order to instruct the system that there can be more valid plans for a single problem. We will clarify this.
>
> Q4. The tokenizer splits predicates and actions into their components: action/predicate name and parameters. E.g. “At Truck1 Loc1” is divided into At, Truck1, and Loc1. These tokens belong to a predefined vocabulary (containing all the action/predicate names and the domain objects); each token is associated to an index of the vocabulary. E.g. for action “Drive Truck1 Loc1 Loc3”,  the tokenizer could give the token sequence [2, 8, 16, 17] where 2 is the index of “Drive” (analogously for 8, 16, 17). This is done for all the GPT input.
>
> Q5. Currently PlanGPT does not prune invalid plans at generation time. Instead, we validate (using VAL) each plan produced by the sampling strategy of PlanGPT and we return the valid one with the lowest number of actions. Incorporating pruning could increase performance. Thank you for the suggestion.

---

### Meta-Review · Area_Chair_6pBQ · 2024-02-03

**Recommendation:** Accept (Oral)
**Confidence:** 4

**Metareview:**

During the discussion period, the reviewers reached a consensus to accept this paper. However, there is also a consensus that the statement "it achieves better or comparable performance than state of the art approaches based on GNN" is a **misrepresentation of the current state-of-the-art and must be removed or amended in the final version**. As pointed out by a reviewer during the discussion, given 10 minutes of evaluation time the GNN approach from (Chen et al, 2024) can solve:

- blocksworld problems of size 75, by training on 40 problems of size 3-10
- visitall problems of size 65, by training on 24 problems of size 3-10
- the number of parameters in their network is on the order of 20K

vs this approach:

- blocksworld problems of size 20 by training on 63,000 problems of size 3-20
- visitall problems of size 10 by training on 63,000 problems of size 2-10
- the network has 83M parameters

We ask the authors to address this issue and follow the other recommendations from the reviewers in the final version of their paper.

(Chen et al, 2024) Learning Domain-Independent Heuristics for Grounded and Lifted Planning. Chen, D., and Thiébaux, S. and Trevizan, F. In Proc. of 38th AAAI Conference on Artificial Intelligence. 2024.

**Ethical Considerations:**

(1) Not Applicable: The paper does not have any ethical considerations to address